# Pre-treatment drug resistance and HIV-1 genetic diversity in the rural and urban settings of Northwest-Cameroon

Joseph Fokam[1,2,3]*, Desire Takou[1], Georges Teto[1], Shu E. Nforbih[2,4], Odine P. Kome[2], Maria M. Santoro[5], Ezechiel S. Ngoufack[1], Mbuh Eyongetah[6], Denis Palmer[7], Estella T. Fokunang[2], Charles N. Fokunang[2], Vittorio Colizzi[1,8,9], Carlo-Federico Perno[1,5,9], Alexis Ndjolo[1,3]

1 Chantal BIYA International Reference Centre for research on HIV/AIDS prevention and management, Yaoundé, Cameroon, 2 Faculty of Medicine and Biomedical Sciences, University of Yaoundé I, Yaoundé, Cameroon, 3 National HIV Drug Resistance Prevention and Surveillance Working Group, Ministry of Public Health, Yaounde, Cameroon, 4 Department of Community Health and Primary Care, University of Lagos, Lagos, Nigeria, 5 University of Rome "Tor Vergata", Rome, Italy, 6 Day Care Center, Bamenda Regional Hospital, Bamenda, Cameroon, 7 Care and Treatment Centre, Mbingo Baptist Hospital, Mbingo, Cameroon, 8 UNESCO Chair of Multidisciplinary Biotechnology and Eurobiopark (Onlus), Rome, Italy, 9 University of Milan, Milan, Italy

* josephfokam@gmail.com

**Data Availability Statement:** Data supporting the findings are available in the results section, and all the 61 sequences generated for this study were submitted to Genbank under the following

## Abstract

### Background

With the scale-up of antiretroviral therapy (ART), pre-treatment drug resistance (PDR) appears $\geq$10% amongst ART-initiators in many developing countries, including Cameroon. Northwest region-Cameroon having the second epidemiological burden of HIV infection, generating data on PDR in these geographical settings, will enhance evidence-based decision-making.

### Objectives

We sought to ascertain levels of PDR and HIV-1 clade dispersal in rural and urban settings, and their potential association with subtype distribution and CD4-staging.

### Methods

A cross-sectional study was conducted from February to May 2017 among patients recently diagnosed with HIV-infection and initiating ART at the Bamenda regional Hospital (urban setting) and the Mbingo Baptist hospital (rural setting). Protease and reverse transcriptase sequencing was performed using an in-house protocol and pre-treatment drug resistance mutations were interpreted using Stanford *HIVdb.v8.3*. Phylogeny was performed for subtype assignation.

### Results

A total of 61 patient sequences were generated from ART initiators (median age: 37 years old; 57.4% female; median CD4 cell count: 184 [IQR: 35–387] in urban vs. 161 [IQR: 96–322] cells/mm³ in rural). Overall, the level of PDR was 9.8% (6/61). Of note, burden of PDR

accession numbers MK995400 - MK995457 and
MK995397 - MK995399.

**Funding:** This study was financially supported by
the Chantal BIYA International Reference Center for
research on HIV/AIDS prevention and
management, in Yaounde, Cameroon. The study
was partly supported by EDCTP2, the European
and Developing Countries Clinical Trial Partnership,
CDF-TMA 1027, which facilitated the relocation at
CIRCB of the principal investigator: Dr. Joseph
Fokam. The funders had no role in study design,
data collection and analysis, decision to publish, or
preparation of the manuscript.

**Competing interests:** The authors have declared
that no competing interests exist.

was almost doubled in urban (12.9% [4/31]) compared to rural setting 6.7% (2/30), $p = 0.352$). Fifteen (15) PDR mutations were found among four patients the urban settings [6 resistance mutations to NRTIs:[M41L (2), E44D (1), K65R (1), K70E (1), M184V/I (2), K219R (1)] and 6 resistance mutations to NNRTIs: K103N (1), E138A/G (2), V179E (1), M230L (1), K238T (1), P225H (1)] against two (02) mutations found in two patients in the rural setting[2 resistant mutations to NNRTIs: E138A (1) and Y188H (1)]. The rural setting showed more genetic diversity (8 subtypes) than the urban setting (5 subtypes), with CRF02_AG being the most prevalent clade (72.1% [44/61]). Of note, level of PDR was similar between patients infected with CRF02_AG and non-CRF02_AG infected (9.1% [4/44]) vs. 11.8% [2/17]), $p = 1.000$). Moreover, PDR appeared higher in patients with CD4 cell count <200 cells/mm$^3$ compared to those with CD4 cell count $\geq$200 cells/mm$^3$ (14.7% [5/34]) vs. 3.7% [1/27]), $p = 0.214$).

## Conclusions

PDR is at a moderate rate in the Northwest region of Cameroon, with higher burden within urban populations. CRF02_AG is the most predominant clade in both urban and rural settings. No effect of HIV molecular epidemiology and CD4-staging on the presence of PDR in patients living in these settings was found. Our findings suggest close monitoring, NNRTI-sparing regimens or sequencing for patients initiating ART, especially in urban settings.

## Introduction

The introduction of highly active antiretroviral therapy (HAART) has significantly decreased the rate of Human Immunodeficiency virus (HIV)–related morbidity and mortality [1]. However, with the new goal currently endorsed by the joint United Nations program on AIDS (UNAIDS) aimed at ending AIDS by 2030, through the 90-90-90 treatment target by 2020 [2,3], and coupled with the WHO recommendation to test and treat [4], there have been an increase in the on-going scale-up of antiretroviral treatment (ART) most especially in sub-Saharan Africa (SSA) [5]. However the increased scale-up has led to undesired consequences like the selection and emergence of HIV drug resistance (HIVDR) variants, which remains a major obstacle, limiting the efficacy of commonly available antiretroviral (ARVs) [6–8]. HIVDR remains a huge problem as it currently involves not only patients on antiretroviral-treatment, but equally drug-naïve individuals carrying HIV-resistant strains [9]. Based on the World Health Organisation (WHO) classification of HIVDR, it is classified into three main categories; pre-treatment drug resistance [PDR, (among ART initiators or re-initiators)], transmitted drug resistance (TDR, among recently-infected individuals), and acquired drug resistance [ADR, (at ART failure)] [10]. ADR occurs when HIV mutations emerge as a result of viral replication in individuals receiving ARV drugs. Transmitted HIV drug resistance (TDR) are resistances detected in ARV drug-naive patients with no history of ARV drug exposure. While pre-treatment HIV drug resistance (PDR) is resistance detected in ARV drug-naive people initiating ART or people with prior ARV drug exposure initiating or reinitiating first-line ART. PDR can either be transmitted or acquired drug resistance. It may have been transmitted at the time of infection (TDR), or it may be acquired by virtue of prior ARV drug exposure [10]. The transmission of HIV-1 drug resistance from treatment-experienced patients to newly infected individuals has been observed in developed countries with access to antiretroviral therapy [11–13]. In 2015, a meta-analysis conducted in sub-Saharan Africa

(SSA), South/Southeast Asia(SSEA), Upper-income Asian countries, Latin America/Caribbean, Europe and North-America showed the median overall prevalence of transmitted HIV drug resistance was 2,8%, 2,9%, 5.6%, 7.6%, 9.4% and 11.5% respectively [14].

Cameroon a sub-Saharan African country situated in Central West Africa, harbours 67% of the Worlds HIV infection [15]. Cameroon endorsed the WHO test and treat recommendation in June 2016 [16]. Before the endorsement, in 2011 the prevalence of HIV/AIDS epidemic for women was almost twice that of men (5.6% Vs 2.9%) with the greatest epidemiological burden in urban settings [17]. Meanwhile ARV drugs through the Ministry of Public Health have been made available free since May 2007 in the National territory of Cameroon for the management of people living with HIV/AIDS (PLHIV) [18,19], with a total of 151 HIV treatment centers all around the national territory by the end of 2011 [19,20]. By 2016, there were 205,359 clients receiving HAART [16], as compared to 145,038 clients in 2014 [21], showing a rapid scale-up of HAART.

Studies carried out in some parts of Cameroon showed a high genetic diversity of HIV with the CRF02_AG being the most predominant form [22]. Several TDR studies conducted in some rural and urban settings of the country found rates varying from low (4.9%) to moderate (9.8%), then to high (24%) levels of drug resistance [21,23–26] with a greater epidemiological burden in the urban than rural setting. In spite of the availability of findings with a global estimate at national level, there is limited evidence on the level of PDR at regional level, and this pitfall could have different implications according to geographical localities. More so, these studies do not focused on assessing HIV-1 genetic diversity and its possible effects on PDR. The Northwest region having the second highest HIV epidemiological burden [17] would require evidence on PDR in both rural and urban settings. Our objectives were; to ascertain PDR in both rural and urban settings in this region, to determine class specific resistance to NRTI, NNRTIs and PIs and its potential association with subtype distribution and CD4 staging for better ART program implementation.

## Materials and methods

### Study design and settings

A descriptive cross-sectional study design was conducted among HIV-infected ART-naïve adults of both sexes enrolled from two clinical study sites; the Bamenda Regional Hospital (an urban setting) and the Mbingo Baptist Hopital-Kom (a rural setting) during a period of 4 months, ranging from February through April 2017.

### Study participants

Eligible participants were those diagnosed HIV-positive, prior to ART initiation, aged 18 years and above, and registered in our study sites.

The sample size of study participants was estimated assuming a 6% error margin and a 95% CI with a PDR rate of 7.55% as earlier described [5], using the following formula:

$$N = \frac{Z^2 \; x \; (p)(1-p)}{d^2}$$

$$Where: \begin{cases} \boldsymbol{N} - the \; minimum \; sample \; size \\ \\ \boldsymbol{Z} - 1.96 \; for \; a \; 95\% \; Confidence \; interval \\ \\ \boldsymbol{p} - prevalence \; of \; patients \; presenting \; with \; resistant \; strain; \; \boldsymbol{p} = 0.075 \; [ref \; 5] \\ \\ \boldsymbol{d} - error \; margin \; set \; at \; 6\% \end{cases}$$

A total sample size of 68 was obtained after calculation. Seventy (70) treatment naïve patients were recruited, of these 63 samples amplified after RT-PCR and 61 sequences were retained for analysis after sequencing reaction. These patients were evenly distributed among the two selected ART facilities (One urban and one rural) The clinical study sites were selected following a community mapping done during former HIV/AIDs intervention programs in the North West region of Cameroon. With the approval of the HIV/AIDs center coordinators in each selected facility, case folders were selected and with the help of the psychosocial workers and community relay agents, the study participants were identified. Advocacy counselling to the selected participants were made and the participants were then included in the study.

## Sample collection

Following the sampling frame, two health facilities (one rural and one urban) were selected by simple random sampling method using balloting procedure. Hospital case files were selected by simple random sampling method using balloting procedures. The units of selection were positively diagnosed HIV naïve participants. The participants from each selected case file was then interviewed and if the respondent was below 18 years or refused to participate the next case file was then chosen. A pre-tested, structured, interviewer-administered questionnaire with open and closed ended questions was used for collecting the respondent's responses. The questionnaire consisted of the socio-demographic and clinical characteristics of the respondents. The socio-demographic parameters consisted of the participants age and sex while the clinical characteristics captured 3 sub-sections; The WHO clinical staging, the immunological status of the participants at time of sample collection (CD4 count) and the third subsection classified the mutations based on the Stanford database list for relevant mutations against NRTIs, NNRTIs and PI. on the antiretroviral drug class identified. A pre-testing of the questionnaire was done in two other selected health facilities each from a rural and urban setting Eight millilitres of whole blood was collected by a trained staff into EDTA blood collection tubes and four (04) plasma aliquots each of 1ml were constituted from every blood sample by centrifugation at 1800 rpm for 10mins and stored at -80˚C. At the end of the collection process, the samples were transported with the help ice packs coolants within a period of 5 hours to the "Chantal BIYA International Reference Centre for research on HIV/AIDS prevention and management" (CIRCB) where they were stored at $-80^0$ C for molecular analysis and bioinformatics.

## Sequencing procedure

Viral RNA was extracted from 1ml aliquot of plasma using the *Purelink™ Viral RNA/DNA kit* following the protocol specifications. A total of 10μl RNA was extracted and was used as a template to amplify the reverse transcriptase and protease genes of HIV-1.

RNA extracts from the patients were amplified using a previously validated in-house reverse transcriptase polymerase chain reaction [27].

Amplified samples from the *pol* region were completely sequenced in the sense and antisense orientations using an automated sequencer (ABI 3130 Genetic Analyzer) with the help of seven different overlapping sequence-specific primers: 5' AGC AGA CCA GAG CCA ACA GC 3' (2140–2159 gag), 5' CCA TCC ATT CCT GGC TTT AAT 3' (2582–2602 pol), 5' CAG GAA TGG ATG GCC CAA AA 3' (2590–2609 pol), 5' TTG TAC AGA AAT GGA AAA GGA AGG 3' (2660–2683 pol), 5' CCC TGT GGA AAG CAC ATT GTA 3' (2985–3004, with an insertion), 5' GCT TCC ACA GGG ATG GAA A 3' (2993–3011 pol), 5' CTA TTA AGT CTT TTG ATG GGT CA 3' (3506–3528 pol) [reference from the HXB2 strain from the Los Alamos National Laboratory database using the Sequence Locator

and QuickAlign tools: http://www.hiv.lanl.gov/content/sequence/HIV/mainpage.html]. The sequencing primers were designed based on the analysis of highly conserved regions among HIV-1 subtypes. The reaction mixture for the sequencing reaction contained 8 µl ABI PRISM Big Dye Terminator (Perkin-Elmer), 4.8 µl water, 3.2 µl primer (1 pmol) and 4 µl of purified cDNA (40 ng), for a total volume of 20 µl. The sequencing conditions were as follows: 35 cycles (96˚C, 10 s; 55˚C, 10 s; 60˚C, 4 min); 1 cycle of 4˚C for 30 min. The quality of each sequence was ensured by covering the PR-RT region with at least two sequence segments (one forward and one reverse). The sequencing product was purified using Sephadex G-50 resin (Sigma-Aldrich) in order to eliminate excess primers, unincorporated dideoxynucleotides (ddNTPs), and salts.

Sequences obtained were aligned in BioEdit Sequence Alignment Editor Version 7.2.6 using CLUSTAL W [28] and compared with reference sequences of all known HIV-1 group M subtypes (A1, A2, B, C, D, F1, F2, G, H, J, and K). Circulating recombinant forms (CRFs) were downloaded from the Los Alamos HIV sequence database (www.hiv-web.lanl.gov). The Phylogenetic tree was constructed using the MEGA version 7 software package with a bootstrap robustness of 1,000 replicates (http://megasoftware.net). Sequences with <70% homology were confirmed using the softwares RDP (Recombinant Detection Programme) and SplitsTree 4 version 4.14.4.

The amplified reverse transcriptase DNA sequences were analysed for potential drug resistance mutations using Stanford HIVdb algorithms (http://hivdb.stanford.edu/) version 8.5, considering possible revertant mutants as sentinel of minority variants. Based on a genotypic susceptibility score, drug resistant mutations detected were classified as high, intermediate or low-level resistance to NRTIs, NNRTIs or PIs.

All the 61 sequences generated for this study were submitted to Genbank using the accession numbers MK995400—MK995457 and MK995397—MK995399. For bivariate analysis, Fischer's exact test was used to compare differences between proportions. Significance level was set at $p < 0.05$

## Ethical considerations

Administrative approval for the study was issued by the Regional Delegation of Public Health of the Northwest region and from the Chantal BIYA International Reference Centre for research on HIV/AIDS prevention and management (Ref. N˚ 2112/016L/CIRCB/DIR/CL); ethical approval for the research was obtained from the Cameroon National Ethics Committee for Research on Human Health (Ref. N˚ 2017/03/893/L/CNERSH/SP), the Institutional Review Board (IRB) of Faculty of Medicine and Biomedical Sciences of the University of Yaoundé I (Ref. N˚ 211/UYI/FMSB/VDRC/CSD) and of the IRB of the CBC Health facility (IRB2017-02). Following study information to each respondent, a written informed consent was obtained after having been clearly intimated with the objectives, methodology, advantages and potential risks involved in the study. Participation was voluntary throughout the study and any participant was free to decline participating in the study at any time without any prejudice. Participants were treated equally irrespective of their social status and other related status. Confidentiality and privacy were ensured by using no identification information. Genotypic results were freely returned to participants for possible clinical benefits in their therapeutic management.

## Results

### Characteristics of the study population

The socio-demographic and clinical characteristics of the successively sequenced treatment naïve patients' samples from rural (n = 30) and urban (n = 31) settings in Northwest Cameroon are recorded in Table 1.

**Table 1. Demographic and clinical characteristics of treatment naïve patients (n = 61) recruited between February-April 2017.**

| Characteristics | Patients (n = 61) |
|---|---|
| Sex: | |
| Women, n (%) | 35 (57.4) |
| Men, n (%) | 26 (42.6) |
| Median age, years (IQR) | 37 (29.5–43) |
| Median CD4 cell count, cells/µl (IQR): | |
| Rural setting | 161 (96–322) |
| Urban setting | 184 (35–387) |

IQR: Interquartile range.

The participants median age was 37 years [IQR: 29.5–43] and 57.4% (35/61) of the study participants were female; Female to male ratio of 1.3:1. At the point of study, the overall study participants in both settings showed an advanced stage of disease: Median CD4, Rural; 161 cells/mm$^3$ [IQR: 96–322], Urban; 184 cells/mm$^3$ [IQR: 35–387].

## Prevalence of HIV-1 drug resistance

In the rural setting 30/31 protease-reverse transcriptase samples were successively genotyped as opposed to 31/32 in the urban setting. This gave a sequencing performance of 96.8% in both settings. The pre-treatment drug resistance (PDR) prevalence was found to be 9.8% (6/61) harbouring at least one drug resistance-associated mutation, with a slightly greater percentage of resistance mutations to Non-nucleotide Reverse transcriptase Inhibitors [NNRTIs] (8.2%) against Nucleotide Reverse transcriptase Inhibitors [NRTIs] (4.9%). Four patients in the urban setting (12.9%) had at least one drug resistance while 2 patients in the rural setting (6.7%) had at least one drug resistance. The patients with PDR was almost doubled in the urban setting [12.9% (4/31)] as compared to the rural setting [6.7% (2/30)]; $p$ = 0.352. In four (4) of the patients carrying resistance associated mutations in the urban setting, 6 resistance mutations to NRTIs: [M41L (2), E44D (1), K65R (1), K70E (1), M184V/I (2), K219R (1)] and 6 resistance mutations to NNRTIs: K103N (1), E138A/G (2), V179E (1), M230L (1), K238T (1), P225H (1)] were found. In the two patients carrying resistance mutations in the samples from the rural setting, resistance mutations to only NNRTIs E138A (1) and Y188H (1)] were found. No mutations conferring resistance to Protease Inhibitors [PIs] were found in our study.

## Subtypes of HIV-1 protease -reverse transcriptase sequences

The samples were obtained from patients who were infected with HIV-1 virus. Overall, CRF02_AG was the most predominant clade [72.2% (44/61)] with similar rates found between urban [74.2% (23/31)] vs rural [70.0% (21/30)] setting, *p-value* = 0.715.

Despite similar rates of CRF02_AG found in both settings, the rural setting showed two more genetic variant [8 subtypes; A$_1$ (3), G (1), F$_2$ (1), CRF02_AG (21), CRF06_cpx (1), CRF11_cpx (1), and CRF18_cpx (1) and an unclassified recombinant of A$_1$/F$_2$ (1)], compared to the Urban setting [5 subtypes A$_1$ (3), G (3), F$_2$ (1), CRF02_AG (23) and CRF18_cpx (1)]. The prevalence Fig 1 shows the phylogenetic tree of the entire study sequence analysed. However the study found no association between subtype distribution (CRF02_AG vs Non-CRF02_AG) and the occurrence of PDR mutation [CRF02_AG [9.1% (4/44)], Non-CRF02_AG [11.8% (2/17)]; $p$ = 1.000], as shown in Table 2. Moreover patients with CD4<200 cells/mm$^3$ were found to have a higher PDR prevalence rate [14.7% (5/34)] as compared to

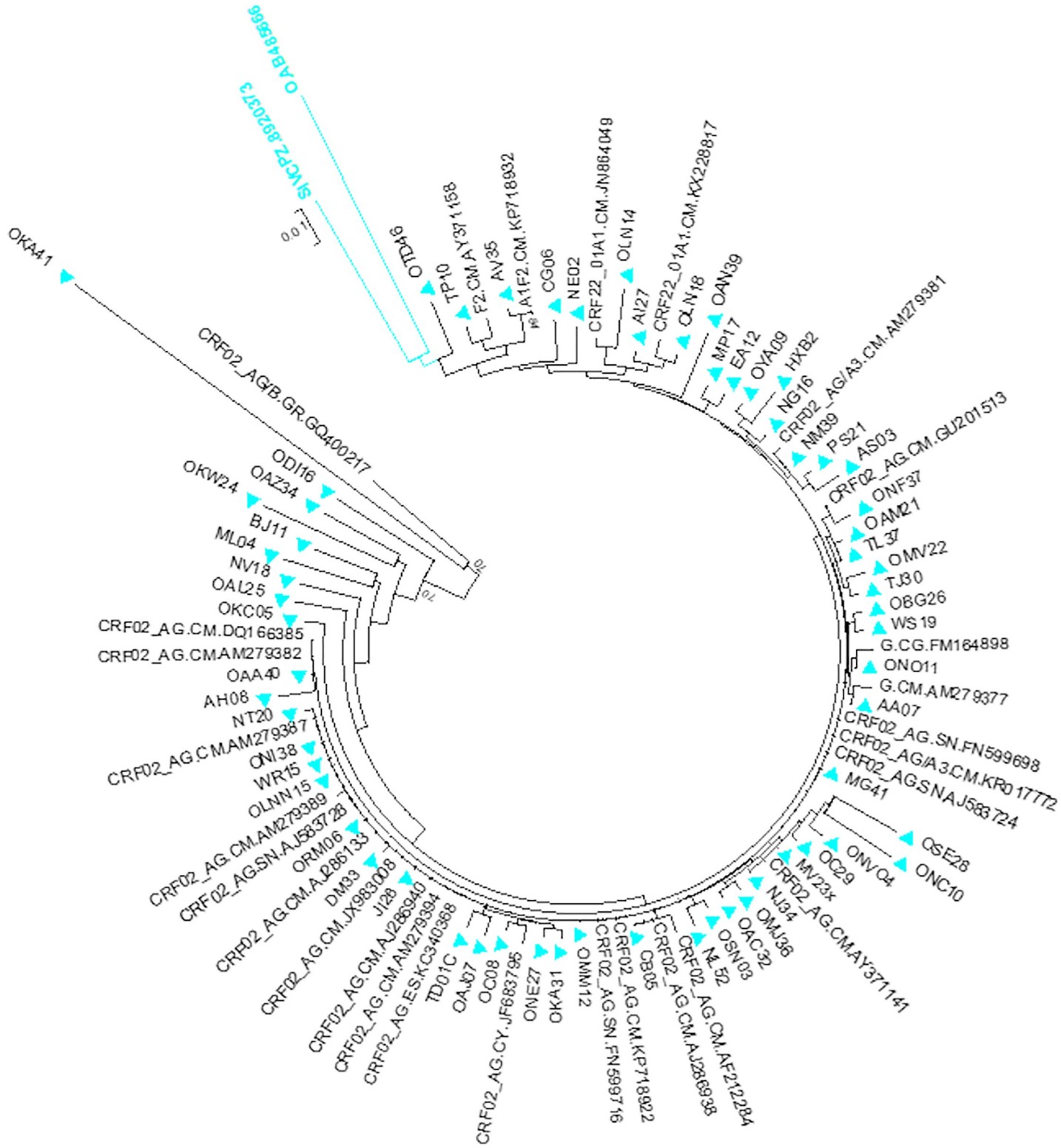

**Fig 1. Phylogenetic tree of HIV-1 isolates from patients in the Northwest region of Cameroon, aligned using Clustal W, analysis was performed using the neighbour-joining method and MEGA 5.0.**

**Table 2. Distribution of drug resistance by HIV-1 clade and WHO CD4 classification.**

| Clade | Presence of DR (n) | Absence of DR (n) | Total (DRM %) |
|---|---|---|---|
| CRF02_AG | 4 | 42 | 44 (9.1) |
| Non-CRF02_AG | 2 | 15 | 17 (11.8) |
| | P-value = 1.000 | | |
| **CD4 Count** | **Presence of DR (n)** | **Absence of DR (n)** | **Total (DRM %)** |
| CD4 < 200 cells/mm$^3$ | 5 | 29 | 34 (14.7) |
| CD4 ≥ 200 cells/mm$^3$ | 1 | 26 | 27 (3.7) |
| | p = 0.214 | | |

DRM: Drug resistance mutation; DR: Drug resistance.

those with CD4 ≥ 200 cells/mm$^3$, however there was no significant association between CD4 staging and the occurrence of drug resistant mutation [3.7% (1/27)]; $p = 0.214$ (Table 2)

Detailed characteristics of patients harbouring pre-treatment drug resistance has been provided in Table 3, considering their geographical locations, their CD4-count level, their HIV-1 subtype, and the patterns of resistance mutations detected.

## Discussion

With the increase risk of emerging drug resistance viruses in developing countries, HIV drug resistance (HIVDR) monitoring is therefore a capital component to support the adherence of patients and consequently pragmatic interventions with respect to the first-line ARV regimens. In this study, the Pre-treatment (PDR) prevalence rate was found to be 9.8%. Keeping in mind that the new surveillance method revised in 2014 by the WHO to monitor PDR in populations initiating ART involves generating a national representative prevalence estimate among this population [29], our findings suggest that PDR is at a moderate level at these settings, in accordance with previously described PDR WHO survey method of populations within a specified geographic area [30]. Our results are in accordance with previous studies carried out in the centre region Yaounde-Cameroon reporting drug-resistance mutation in population initiating ARV ranging from 4.9% to 24% [21,23–26]. A recent study characterising a national representation of PDR in Cameroon following recent guidelines by WHO showed PDR resistance in Cameroon to be 10.4% [31]. This would suggest that though standard first-line treatment may remain effective, their efficacy is been affected as rate of PDR is on the rise. Studies conducted in other African countries show drug resistance mutation in populations initiating ARV to be; 11.5% (Mali), 9.2% (South Africa), 9.2% (Kenya), 4.2% (Morocco) [32–35]. In Europe precisely in the UK PDR has been found to reach right up to 14% [36]. Rates as high as 25% of PDR has been reported in the USA [37]. Of note most of these studies accessed transmitted drug

**Table 3. Drug resistance mutation patterns.**

| Patient ID | CD4 count (cells/mm3) | Drug resistance mutations | | | Subtype | ART facility setting |
|---|---|---|---|---|---|---|
| | | **NRTIs** | **NNRTIs** | **PIs** | | |
| ORM06_CIRCB_2017 | 43 | M41L, E44EAD | - | - | CRF02_AG | Urban |
| OSN03_CIRCB_2017 | 310 | K65R, M184I | V179E, M230L | - | CRF02_AG | Urban |
| OAM21_CIRCB_2017 | 1 | M41L, K70E, M184V, K219R | K103N, E138G, P225H, K238T | - | CRF02_AG | Urban |
| OAZ34_CIRCB_2017 | 29 | - | E138A | - | G | Urban |
| BJ11_CIRCB_2017 | 161 | - | Y188YH | | CRF02_AG | Rural |
| TJ30_CIRCB_2017 | 71 | - | E138A | | CRF02_AG | Rural |

resistance in naïve patients as distinguished now from pre-treatment drug resistance as per WHO revision in 2014 [29]. The differences in prevalence of PDR can be due to several factors; among which we could have the nature of the study (cross-sectional versus longitudinal), as longitudinal studies has been shown to have a lower level of drug resistance in naïve population compared to cross-sectional [8]. Secondly, differences in data among countries may also have resulted from specific biological and cultural characteristics as they relate to national epidemics, such as transmission routes, the proportion of non-B viruses and prescription guidelines [8]. Finally discrepancies can be further explained by the fact that, there are different algorithms available to the public domain which are routinely used for drug resistance interpretation and can therefore give results which are more or less similar, depending on the capacity of these tools to detect mutations [5]. However, the Stanford HIVdb algorithm version 8.5 known for its high sensitivity in the HIVDR detection was used in our study.

Our study showed a majority of mutations to NNRTIs (about two times higher than resistance mutations to NRTIs). There were no mutations to PIs detected in our study. This can be explained by the fact that majority of the population under HAART are not exposed to protease inhibitors which are generally reserved for second line treatment in our setting. Therefore the rate of drug resistance mutations conferring resistance to PI among populations initiating ARV is minimal. This is in accordance with a previous study which showed that only secondary mutations associated with PIs were detected in drug naïve patients in Cameroon during 2000 to 2002 [38]. However, another study in 2006 detected primary resistance to PI among naïve patients in Cameroon, revealing a rate as high as 7.4% [39]. This could be explained by the fact that the study had a larger sample size and the utilisation of deep sequencing method was employed which could detect minor mutations conferring resistance to PIs. Of note patients in the rural setting presented with resistance only to the NNRTIs in our study. This is in conformity of a recent study in Cameroon that showed presence of mutation to NNRTIs among naïve patients in rural areas only, with prevalence of PDR to NNRTI being 4.3% [31]. Another study in rural Cameroon showed PDR among population initiating ARV to be 4.8% [40]. The low prevalence experienced in rural areas could be explained by the fact that these areas have not been exposed to ARV for a very long period as compared to urban areas where ARV have been the main stay. The low genetic variability of these drugs can also influence their apparition. However there have been an increase in trend of PDR amongst populations initiating ARV in Cameroon from 0% in 1996–99 to 12.3% in 2007 [40] with this increase experienced more in NNRTIs. A recent retrospective study by Hassan et al in Kenya [41] using next generation sequencing (NGS) showed 24% of participants had at least one PDR variant with 8% to NRTIs, 6% to NNRTIs and 12% to PIs. The higher PDR found in their study when compared to ours is explained by the fact they used a more sensitive sequencing technique which detects mutation variants which cannot be detected using the standard sanger sequencing method. However the prevalence of NRTIs and NNRTIs is similar to that of our study. Similarly, a study conducted by Mbunkah *et al* in Cameroon [42] using NGS on dried blood spots found a low prevalence of transmitted drug resistance most especially in PIs, as fewer patients (<5%) are exposed to these regimens in the country [5,42]. Therefore, efforts for continuous surveillance monitoring for PDR should be intensified with NGS emphasis laid on rural areas that would be much more exposed to HIV drug-resistance as ART scale-up increases.

The mutation E138A found both in rural and urban setting confers low-level resistance to second generation NNRTIs and their presence alone or in conjunction with the mutation M138I causes decreased susceptibility of the virus to ETV and RPV [43]. This mutation has been found to occur between 0.5 to 5% among naïve patients [44]. The mutation Y188YH

(quasi-specie) found only in the rural setting renders an intermediate level of resistance to the first-line NNRTIs [45].

In the urban setting, mutations conferring resistance to both NRTIs and NNRTIs but not PIs were found. The most common mutation to NRTIs found in our study was M41L (13.33%, 2/15). The mutation M41L is a TAM that usually occurs with T215Y. In combination, M41L plus T215Y confer intermediate / high-level resistance to Zidovudine (AZT) and Stavudine (d4T) and contribute to reduced Didanosine (ddI), Abacavir (ABC) and Tenofovir (TDF) susceptibility [46]. Of note the mutation K65R was found in one naïve patient. This mutation has mostly been described in patients under HAART but has hardly been found in patients' naïve to ARV therapy [47]. K65R causes intermediate/high-level resistance to TDF, ddI, ABC and d4T and low/intermediate resistance to Lamivudine (3TC) and Emtricitabine (FTC) but has also been found to increase susceptibility to AZT [48]. The emergence of K65R in naïve patients in our study strongly suggest reduced efficacy of first line regimens, hence the need for proper surveillance of this mutation among this population. The mutation M184V which on its own confers high level resistance to 3TC and FTC (about 100-fold increase) [49] was detected amongst these patients. The presence of this mutation could be explained by the fact 3TC is one of the principal drugs used in first-line therapy in Cameroon and equally constitute the backbone in the PMTCT of HIV. However, TDF in the presence of 3TC has been shown to induce M184V mutation which enhances the efficacy of 3TC-TDF combination [48]. The mutations k103N, E138G, P225H. K238T, V179E, M230L and E138A found in the participants in the urban setting conferred resistance to NNRTIs. K103N and M230L conferred high level resistance to the first-line NNRTIs. These findings are similar to that reported in a similar study in an urban setting in Cameroon [50]. The presence of these mutation is understood because Efavirenz EFV and Nevirapine (NVP) are molecules widely used in the first-line regimen and in addition nevirapine is the key antiretroviral for the PMTCT in Cameroon.

As pertaining to HIV-1 genetic variability, the phylogenetic analysis of the protease-reverse transcriptase *pol* region of the sequences in both settings revealed a marked genetic diversity most predominantly in the rural setting than urban. The CRF02_AG was the most represented strain in this region, in accordance with studies in other regions of Cameroon showing the prevalence of this strain [19,50]. It has been shown that the CRF02_AG strain have a biologic advantage over parental strains, including a possibly higher replicative fitness and/or transmission capacity [51]. In addition the founder effect has also been argued for the predominance of CRF02_AG in West Central Africa which stipulates that the recombinant strain first introduced in a particular area, consequently get established in a population before other subtypes enter into the scene [51]. These could explain the predominance of the CRF02_AG strain in our study.

Finally, in both settings we found no association between CD4 staging (immune status), subtype distribution and the occurrence of drug resistance mutation. Our results are in accordance with a similar study done in the Centre region (Yaounde) Cameroon in 2014 [5]. However further studies with larger sample size are recommended to better affirm our findings.

## Study limitations

This study had some limitations which include, the fact that we used conventional Sanger sequencing which does not detect minor virus populations if frequencies are less than 20–25% of the entire viral population found within an infected individual. Therefore, minor mutations that have been shown to affect ARV drug susceptibility could not have been detected following our method. Furthermore, the use of deep sequencing for full length would have help in delineating potential new viral strain herein reported as unclassified. Also, the small sample size in

our study could have an effect on the overall estimates of PDR; another limitation is our criteria for classifying drug naïve individual based solely on the testimonies of our patients, as some could have information bias about their status, thus supporting the measurement of drug concentration in future studies. The lack of viral load testing also limited the ability in assessing the effect of plasma viral load on PDR. This concept could now be easily addressed as viral load as become free of charge in the Cameroon ART program since January 2020.

## Conclusion

PDR was at a moderate rate in the two settings and higher within the urban populations. Though CRF02_AG is predominant in both urban and rural setting, viral diversity was found to be higher within the rural population. We found no association between HIV subtypes, CD4 staging and the occurrence of PDR mutation amongst these patients. Our findings suggest that there is need for close virological monitoring, use of NNRTI-sparing regimens, or HIV-1 genotyping for patients initiating ART in urban settings, especially with the presence of K65R mutation that jeopardizes the effectiveness of the current preferred first line ART.

## Acknowledgments

We are thankful to patients who provided their consent for enrolment, and to administrative, clinical and technical staffs of the study sites for facilitating study implementation. The present study was conducted within the frame of research theses of Dr Shu Emile Nforbih and Dr Odine Padimel Kome at the Faculty of Medicine and Biomedical Sciences of the University of Yaounde I.

## Author Contributions

**Conceptualization:** Joseph Fokam, Shu E. Nforbih, Maria M. Santoro, Denis Palmer, Estella T. Fokunang, Charles N. Fokunang, Vittorio Colizzi, Carlo-Federico Perno, Alexis Ndjolo.

**Data curation:** Desire Takou, Odine P. Kome, Ezechiel S. Ngoufack.

**Formal analysis:** Georges Teto, Mbuh Eyongetah.

**Investigation:** Joseph Fokam, Shu E. Nforbih, Odine P. Kome, Denis Palmer, Alexis Ndjolo.

**Methodology:** Joseph Fokam, Desire Takou, Georges Teto, Shu E. Nforbih, Odine P. Kome, Maria M. Santoro, Ezechiel S. Ngoufack, Mbuh Eyongetah, Charles N. Fokunang.

**Project administration:** Joseph Fokam, Alexis Ndjolo.

**Supervision:** Desire Takou, Maria M. Santoro, Estella T. Fokunang, Charles N. Fokunang, Vittorio Colizzi, Carlo-Federico Perno, Alexis Ndjolo.

**Validation:** Joseph Fokam, Shu E. Nforbih, Odine P. Kome, Mbuh Eyongetah, Denis Palmer, Estella T. Fokunang.

**Visualization:** Ezechiel S. Ngoufack.

**Writing – original draft:** Joseph Fokam, Shu E. Nforbih, Odine P. Kome, Mbuh Eyongetah.

**Writing – review & editing:** Joseph Fokam, Desire Takou, Georges Teto, Maria M. Santoro, Ezechiel S. Ngoufack, Denis Palmer, Estella T. Fokunang, Charles N. Fokunang, Vittorio Colizzi, Carlo-Federico Perno, Alexis Ndjolo.

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
