## [Decision Letter · Decision Letter 0]

15 Apr 2020

PONE-D-19-27597

Pre-treatment Drug Resistance and HIV-1 Genetic Diversity in the Rural and Urban Settings of Northwest-Cameroon

PLOS ONE

Dear Dr Fokam,

Thank you for submitting your manuscript to PLOS ONE. After careful consideration, we feel that it has merit but does not fully meet PLOS ONE’s publication criteria as it currently stands. Therefore, we invite you to submit a revised version of the manuscript that addresses the points raised during the review process.

We would appreciate receiving your revised manuscript by 60 days. To enhance the reproducibility of your results, we recommend that if applicable you deposit your laboratory protocols in protocols.io, where a protocol can be assigned its own identifier (DOI) such that it can be cited independently in the future. For instructions see: http://journals.plos.org/plosone/s/submission-guidelines#loc-laboratory-protocols

We look forward to receiving your revised manuscript.

Kind regards,

Francesca Ceccherini-Silberstein, Ph.D.

Academic Editor

PLOS ONE

Journal Requirements:

2. Please provide additional details regarding participant consent. In the ethics statement in the Methods and online submission information, please ensure that you have specified what type you obtained (for instance, written or verbal, and if verbal, how it was documented and witnessed) - "written/verbal" is not sufficient.

3. You state  "sample size of 61 participants was calculated" with no description of any sample size calculation/the associated parameters.

4. Please include additional information regarding the survey or questionnaire used in the study and ensure that you have provided sufficient details that others could replicate the analyses. For instance, if you developed a questionnaire as part of this study and it is not under a copyright more restrictive than CC-BY, please include a copy, in both the original language and English, as Supporting Information. This includes the nature and number of participants included in the pretesting of this questionnaire.

5. "We noticed you have some minor occurrence of overlapping text with the following previous publication(s), which needs to be addressed:

http://natap.org/2005/HIV/090205_10.htm

https://www.liebertpub.com/doi/10.1089/aid.2007.0156

https://apps.who.int/iris/bitstream/handle/10665/255896/9789241512831-eng.pdf;jsessionid=22C33E7D74568025240C4E184EE267BC?sequence=1

In your revision ensure you cite all your sources (including your own works), and quote or rephrase any duplicated text outside the methods section. Further consideration is dependent on these concerns being addressed.

6. Thank you for stating the following financial disclosure:

"The funders had no role in study design, data collection and analysis, decision to publish, or preparation of the manuscript"

7. Thank you for stating the following in your Competing Interests section: "None"

Reviewers' comments:

Reviewer's Responses to Questions

**Comments to the Author**

1. Is the manuscript technically sound, and do the data support the conclusions?

Reviewer #1: Yes

Reviewer #2: Yes

2. Has the statistical analysis been performed appropriately and rigorously? 

Reviewer #1: Yes

Reviewer #2: Yes

3. Have the authors made all data underlying the findings in their manuscript fully available?

Reviewer #1: Yes

Reviewer #2: Yes

4. Is the manuscript presented in an intelligible fashion and written in standard English?

Reviewer #1: Yes

Reviewer #2: Yes

5. Review Comments to the Author

Reviewer #1: The manuscript provides information about the prevalence of HIV drug resistance in a small population of Cameroonian drug-naïve patients enrolled just in 4 months. In addition, the authors tried to find potential correlations between drug resistance and viral load or CD4 cell count, but due to the low number of enrolled individuals, no evidence for that was found. Thus, valuating the clinical impact of drug resistance in this kind of population remains challenging. In order to have a real overview of drug resistance in urban and rural area, and to really evaluate its clinical implication, more patients should be tested in a longer time period. This is the most important limitation for this study, that should be accurately stressed along all the manuscript.

Below are major comments:

1) The authors asserted to have considered for their study pre-treatment HIV drug resistance (PDR) mutations, defined like transmitted or acquired drug resistance. According to what the authors wrote, these mutations may have been transmitted at the time of infection (TDR), or it may be acquired by prior ARV drug exposure. However, which mutations the authors used for their analysis remain unclear. A list with all the NRTI, NNRTI, and PI resistance mutations used for this study should be added in the supplementary material. For example, were the revertant mutations at RT position 215 considered in this list? If not, these mutations should be added for their role in drug resistance. These revertants might easily develop in T215Y/F and their presence by standard sequencing may indicate the presence of T215Y/F as a minority variant.

2) Results, “Prevalence of HIV-1 Drug Resistance” paragraph. The authors stated that the PDR prevalence in rural and urban setting is 12.9% and 6.7%, respectively. It is not clear if this prevalence reflects the number of patients carrying at least one drug resistance. Please, clarify. The authors should also define clearly the number of patients carrying at least one NRTI and NNRTI mutation.

3) In order to better evaluate drug resistance mutation patterns, the authors should add a table that summarizes for each one of the 6 patients carrying drug resistance, viral load, CD4, drug resistance mutations, subtype, and location of ART facility where the diagnosis was done (urban or rural).

4) Discussion section. The authors stated that their study “showed a majority of mutations to NNRTIs (52.94%) than NRTIs (47.06%)”. The prevalence of NRTI and NNRTI resistance should be reported on the overall population, and not, as done, on the 6 patients infected by drug resistant virus.

5) Discussion section. The authors should compare their results with the most recent literature published on this field in Africa. In a recent paper based on NGS, Hassan et al defined the prevalence of drug resistance mutations in HIV-1 infected drug naïve patients from rural area of Kenya (Hassan et al., PloS One 2019). The authors reported a 24.0% of participants with at least one drug resistance, 12% against PI, 8.0% against NRTI and 6% against NNRTI. Despite the use of NGS, the prevalence of NRTI and NNRTI resistance in this paper is quite similar to those reported by Fokam et al. Differently, the prevalence of PI resistance that is zero in Fokam et al. is similar to that reported by another Cameroonian paper using NGS technology (Koizumi, JAC 2006).

Minor revision:

1) Abstract and Results: Please, report drug resistance mutation without quasispecies (i.e. no M41ML but M41L).

2) Results, Subtypes of HIV-1 Protease-Reverse Transcriptase sequences paragraph. Instead of A1/F2 subtype, define the exact recombinant form of this sequence.

3) Results, Subtypes of HIV-1 Protease-Reverse Transcriptase sequences paragraph. Please, revise the sentence “the rural setting showed more two more genetic variant [8 subtypes; A1 (3), G (1), F2 (1), CRF02_AG (21), CRF06_cpx (1), CRF11_cpx (1), CRF18_cpx (1) and A1/F2 (1)], compared to the Urban setting [5 subtypes A1 (3), G (3), F2 (1), CRF02_AG (23) and CRF18_cpx (1)].” What does “more two more genetic variant” mean? Add the prevalence of subtypes in rural and urban settings, and p-value.

4) Figure 1, resolution should be improved

5) Please, revise reference 10. The HIV drug resistance report 2017 is available at the following link: https://www.who.int/hiv/pub/drugresistance/hivdr-report-2017/en/

6) Authors used PDR or DRM interchangeably. Please, use an uniform terminology along all the manuscript.

Reviewer #2: This work quantifies the prevalence of pre-treatment drug resistance (PDR) in both rural and urban settings in Cameroon. A total of 61 sequences from 61 patients were analysed for NNRTIs and NRTIs resistance mutations. Fifteen resistance mutations were found in 4 patients from the urban setting compared with only two mutations (NNRTIs only) in two patients from the rural setting. Comparison between rural versus urban setting or level of CD4 are limited by the small sample size. The manuscript needs careful reading to correct few typos or English wording. I have only few minor comments

1. There are only few mutations observed in few patients so the complete list of mutations for the four patients from the urban setting should be given.

2. It is somewhat over-interpreted to consider that the difference in CD4 between urban and rural of 184 vs. 161 indicate a delay in diagnosis. Both IQR are very large likely as the range.

3. Comparing the level of CD4 and the prevalence of mutations shows no statistical difference likely due to the small sample size. I’m surprised that the level of HIV-1 RNA was not available in the study to compare the presence of mutations according to the level of viral load at genotyping time. I guess that the CD4 count is a proxy of the level of viral load based on the correlation between HIV-1 RNA and CD4 cell count. This point should be discussed.

4. In the discussion Section, I’m not convinced that discrepencies between studies can be explained by the different algorithms used. I understood that the percent gives for the studies are % of resistance mutations not resistance to drugs. The difference in the list of mutations used in the different algorithms is marginal.

6. PLOS authors have the option to publish the peer review history of their article (what does this mean?). If published, this will include your full peer review and any attached files.

Reviewer #1: No

Reviewer #2: No

---

## [Author Response · Author response to Decision Letter 0]

18 May 2020

RESPONSE TO COMMENTS

Journal Requirements

Comment #1: Please ensure that your manuscript meets PLOS ONE's style requirements, including those for file naming.

Authors: We thank the academic editor for this comment. We have corrected the whole document to ensure the manuscript follows the PLOS ONE style template found on the shared link.

Comment #2: Please provide additional details regarding participant consent. In the ethics statement in the Methods and online submission information, please ensure that you have specified what type you obtained (for instance, written or verbal, and if verbal, how it was documented and witnessed) - "written/verbal" is not sufficient.

Authors: We thank the academic editor for this pertinent remark. The statement has been revised as follows (see page 8, lines 280-293): 

“Administrative approval for the study was issued by the Regional Delegation of Public Health of the Northwest region and from the Chantal BIYA International Reference Centre for research on HIV/AIDS prevention and management (Ref. N° 2112/016L/CIRCB/DIR/CL); ethical approval for the research was obtained from the Cameroon National Ethics Committee for Research on Human Health (Ref. N° 2017/03/893/L/CNERSH/SP), and the Institutional Review Board (IRB) of Faculty of Medicine and Biomedical Sciences of the University of Yaoundé I (Ref. N° 211/UYI/FMSB/VDRC/CSD). Following study information to each respondent, a written informed consent was obtained after having been clearly intimated with the objectives, methodology, advantages and potential risks involved in the study. Participation was voluntary throughout the study and any participant was free to decline participating in the study at any time without any prejudice. Participants were treated equally irrespective of their social status and other related status. Confidentiality and privacy were ensured by using no identification information. Genotypic results were freely returned to participants for possible clinical benefits in their therapeutic management.” 

Comment #3: You state “sample size of 61 participants was calculated" with no description of any sample size calculation/the associated parameters.

Authors: We thank the academic editor for this remark. The detailed description of the sample size calculation has been provided in page 5, lines 135-143. The formula used is as follows:

N =( Z^2 x (p)(1-p) )/(d² )

 N - the minimum sample size

 Z - 1.96 for a 95 % Confidence interval

Where: p – prevalence of patients presenting with resistant strain; p=7.55% [ref 5] 

 d – error margin set at 6%

The authors considered a 94% CI with an error margin set at 6%. Prevalence of 0.75 from previous studies by Fokam et al in same country was used for the sample size calculation. A sample size of 68 participants was obtained. Seventy treatment naïve patients were recruited, of these 63 samples amplified after RT-PCR and 61 sequences were retained for analysis after sequencing reaction. Analysis was done based on the 61 samples retained.

Comment #4: Please include additional information regarding the survey or questionnaire used in the study and ensure that you have provided sufficient details that others could replicate the analyses.

Authors: we thank the Academic editor for this remark, more clarifications have been made in the manuscript under methodology (sample collection), at page 6, lines 218-223.

Comment #5: We noticed you have some minor occurrence of overlapping text with the following previous publication(s), which needs to be addressed

Authors: We thank the academic editor for this very pertinent remark. We acknowledge some minor overlapping which has been rephrased. 

Comment #6: Thank you for stating the following financial disclosure: "The funders had no role in study design, data collection and analysis, decision to publish, or preparation of the manuscript" Please address the following queries: Please clarify the sources of funding

Authors: We thank the Academic editor for this pertinent comment and remark. We acknowledge this mistake on our part. The source of funding has been included on the section titled ‘Funding” in the manuscript and corresponding disclosure statement stated.

Comment #7: Thank you for stating the following in your Competing Interests section: "None"

Authors: We thank the academic editor for this pertinent remark. These remarks have been taken into account accordingly in the section labeled “Competing interest” in the manuscript. These have equally been addressed on the online completion form.

 

REVIEWER’S COMMENTS TO AUTHOR

Reviewer #1

Major comments:

Comment #1: The authors asserted to have considered for their study pre-treatment HIV drug resistance (PDR) mutations, defined like transmitted or acquired drug resistance. According to what the authors wrote, these mutations may have been transmitted at the time of infection (TDR), or it may be acquired by prior ARV drug exposure. However, which mutations the authors used for their analysis remain unclear. A list with all the NRTI, NNRTI, and PI resistance mutations used for this study should be added in the supplementary material. For example, were the revertant mutations at RT position 215 considered in this list? If not, these mutations should be added for their role in drug resistance. These revertants might easily develop in T215Y/F and their presence by standard sequencing may indicate the presence of T215Y/F as a minority variant.

Authors: We thank the reviewer #1 for this very pertinent comment. As stated in the methodology section; paragraph 5 of analysis, analysis for drug resistance mutations was done using the HIVdb algorithm (http://hivdb.standford.edu/) version 8.5. Following the high precision remark on the revertant mutations at RT position 215, indeed it was considered in the list of mutations from Stanford HIVdb used for analysis. None of the patients had these mutations. A list with all the NRTI, NNRTI and PI resistance mutations has been added in the supplementary material as stated by the reviewer #1. See page 7, lines 266-270.

Comment #2: Results, “Prevalence of HIV-1 Drug Resistance” paragraph. The authors stated that the PDR prevalence in rural and urban setting is 12.9% and 6.7%, respectively. It is not clear if this prevalence reflects the number of patients carrying at least one drug resistance. Please, clarify. The authors should also define clearly the number of patients carrying at least one NRTI and NNRTI mutation. 

Authors: we thank the reviewer #1 for this remark, the comment is very pertinent. Of the 6 patients who had drug resistances in both settings, 4 patients in the urban setting had at least one drug resistance (12.9%) while 2 patients (6.7%) had at least one drug resistance in the rural setting. The statement has been rephrased for better clarification on the identified section on the manuscripts. See details in pages 9-10, lines 335-339.

Comment #3: In order to better evaluate drug resistance mutation patterns, the authors should add a table that summarizes for each one of the 6 patients carrying drug resistance, viral load, CD4, drug resistance mutations, subtype, and location of ART facility where the diagnosis was done (urban or rural).

Authors: we thank the reviewer #1 for this remark, the comment is very pertinent. The summarized table 3 has been added accordingly. See pages 11-12, lines 389-393.

Comment #4: Discussion section. The authors stated that their study “showed a majority of mutations to NNRTIs (52.94%) than NRTIs (47.06%)”. The prevalence of NRTI and NNRTI resistance should be reported on the overall population, and not, as done, on the 6 patients infected by drug resistant virus.

Authors: we thank the reviewer #1 for this pertinent remark showing the detailed reading of the submission package. Appropriate corrections has been made accordingly on paragraph 2, discussion section. See page 13, lines 425-426.

Comment #5: Discussion section. The authors should compare their results with the most recent literature published on this field in Africa. In a recent paper based on NGS, Hassan et al defined the prevalence of drug resistance mutations in HIV-1 infected drug naïve patients from rural area of Kenya (Hassan et al., PloS One 2019). The authors reported a 24.0% of participants with at least one drug resistance, 12% against PI, 8.0% against NRTI and 6% against NNRTI. Despite the use of NGS, the prevalence of NRTI and NNRTI resistance in this paper is quite similar to those reported by Fokam et al. Differently, the prevalence of PI resistance that is zero in Fokam et al. is similar to that reported by another Cameroonian paper using NGS technology (Koizumi, JAC 2006).

Authors: we thank the reviewer #1 for this pertinent update. The respective manuscripts have been read and used in our discussion section. See page 13, lines 444-452.

Minor comments:

Minor comment #1: Abstract and Results: Please, report drug resistance mutation without quasi species (i.e. no M41ML but M41L).

Authors: we thank the reviewer #1 for this remark. The corrections have been taken into account in the manuscript as recommended.

Minor comment #2: Results, Subtypes of HIV-1 Protease-Reverse Transcriptase sequences paragraph. Instead of A1/F2 subtype, define the exact recombinant form of this sequence.

Authors: we thank the reviewer #1 for this comment. On the list of circulating recombinant form (CRF), we did not find any classified CRF with these two distinct subtypes. 

Minor comment #3: Results, Subtypes of HIV-1 Protease-Reverse Transcriptase sequences paragraph. Please, revise the sentence “the rural setting showed more two more genetic variant [8 subtypes; A1 (3), G (1), F2 (1), CRF02_AG (21), CRF06_cpx (1), CRF11_cpx (1), CRF18_cpx (1) and A1/F2 (1)], compared to the Urban setting [5 subtypes A1 (3), G (3), F2 (1), CRF02_AG (23) and CRF18_cpx (1)].” What does “more two more genetic variant” mean? Add the prevalence of subtypes in rural and urban settings, and p-value.

Authors: we thank the reviewer #1 for this comment. The word “more two more genetic variant’ is a duplication which has been corrected. Actually, the statement was aimed at informing the rural setting had two genetic variants which the urban setting didn’t. These variants were CRF06_cpx and CRF11_cpx. The prevalence of subtypes was evaluated based on the clade. Participants were classified as either having CRF02_AG or non CRF02_AG and based on the setting a non-significant p-value of 0.715 was obtained. This is explained in paragraph 1 of “Results, Subtypes of HIV-1 Protease-Reverse Transcriptase sequences paragraph.” See page 10, lines 356-366.

Minor comment #4: Figure 1, resolution should be improved

Authors: we thank the reviewer #1 for this remark. The resolution has been improved upon as can be seen on the attached file.

Minor comment #5: Please, revise reference 10. The HIV drug resistance report 2017 is available at the following link: https://www.who.int/hiv/pub/drugresistance/hivdr-report-2017/en/

Authors: we thank the reviewer #1 for this remark showing the detailed reading of the submission package. The reference has been corrected on the manuscript.

Minor comment #6: Authors used PDR or DRM interchangeably. Please, use an uniform terminology along all the manuscript.

Authors: we thank the reviewer #1 for this remark. The whole document has been reviewed and corrections made accordingly, to read only PDR.

 

Reviewer # 2

Minor comments

Minor comment #1: There are only few mutations observed in few patients so the complete list of mutations for the four patients from the urban setting should be given.

Authors: we thank the reviewer #2 for this suggestion. The suggestion goes in line with a comment raised by reviewer #1. The lists of mutations and related data have been represented on table 3.

Minor comment #2: It is somewhat over-interpreted to consider that the difference in CD4 between urban and rural of 184 vs. 161 indicate a delay in diagnosis. Both IQR are very large likely as the range.

Authors: we thank the reviewer #2 for this pertinent comment. This statement has been removed from the text.

Minor comment #3: Comparing the level of CD4 and the prevalence of mutations shows no statistical difference likely due to the small sample size. I’m surprised that the level of HIV-1 RNA was not available in the study to compare the presence of mutations according to the level of viral load at genotyping time. I guess that the CD4 count is a proxy of the level of viral load based on the correlation between HIV-1 RNA and CD4 cell count. This point should be discussed.

Authors: we thank the reviewer #2 for this very pertinent remark. Indeed, we also had the same need in viral load testing throughout for the study. However, this could not be realized because viral load testing was not recommended at ART initiation and the cost of viral load testing was not available. Of note, at the time of the study, viral loads were still payable in Cameroon, and the Cameroon government has only recently adopted free viral loads since January 2020. This has been highlighted in the discussion section, under study limitation. 

Minor comment #4: In the discussion Section, I’m not convinced that discrepancies between studies can be explained by the different algorithms used. I understood that the percent gives for the studies are % of resistance mutations not resistance to drugs. The difference in the list of mutations used in the different algorithms is marginal.

Authors: we thank the reviewer #2 for this remark. Indeed, resistance mutations are expressed in percentages. Moreover, the sequencing method has a very important role in the detection of the resistance mutations. For example, the sanger sequencing technic used in our study does not detect viruses with major mutations like M46I unlike other WHO major mutations like K103N that can be easily detected because of its high replicative fitness. The sequencing method used can therefore be improved for better identification of mutations if further deep sequencing is performed, which can henceforth explain the difference in the detected mutations.

In a nutshell, we are very appreciative for your informed review and we hope our revised version would meet your approval of the revised manuscript. 

Sincerely, 

Dr. Joseph Fokam

---

## [Decision Letter · Decision Letter 1]

26 Jun 2020

Pre-treatment Drug Resistance and HIV-1 Genetic Diversity in the Rural and Urban Settings of Northwest-Cameroon

PONE-D-19-27597R1

Dear Dr. Fokam,

We’re pleased to inform you that your manuscript has been judged scientifically suitable for publication and will be formally accepted for publication once it meets all outstanding technical requirements.

Kind regards,

Francesca Ceccherini-Silberstein, Ph.D.

Academic Editor

PLOS ONE

Reviewers' comments:

Reviewer #1: All comments have been addressed

Reviewer #2: All comments have been addressed

2. Is the manuscript technically sound, and do the data support the conclusions?

Reviewer #1: Yes

Reviewer #2: Yes

3. Has the statistical analysis been performed appropriately and rigorously? 

Reviewer #1: Yes

Reviewer #2: Yes

4. Have the authors made all data underlying the findings in their manuscript fully available?

Reviewer #1: Yes

Reviewer #2: Yes

5. Is the manuscript presented in an intelligible fashion and written in standard English?

Reviewer #1: Yes

Reviewer #2: Yes

6. Review Comments to the Author

Reviewer #1: (No Response)

Reviewer #2: All the comments have been apropriately adressed.

The modifications of the manuscript are apropriate.

7. PLOS authors have the option to publish the peer review history of their article (what does this mean?). If published, this will include your full peer review and any attached files.

Reviewer #1: No

Reviewer #2: No

---

## [Editor Report · Acceptance letter]

7 Jul 2020

PONE-D-19-27597R1 

Pre-treatment Drug Resistance and HIV-1 Genetic Diversity in the Rural and Urban Settings of Northwest-Cameroon 

Dear Dr. Fokam:

I'm pleased to inform you that your manuscript has been deemed suitable for publication in PLOS ONE. Congratulations! Your manuscript is now with our production department. 

Kind regards, 

on behalf of

Dr. Francesca Ceccherini-Silberstein 

Academic Editor

PLOS ONE